# SEAN: A Simple and Efficient Attention Network for Aircraft Detection in SAR Images

**Ping Han [1,\*], Dayu Liao [1], Binbin Han [1] and Zheng Cheng [2]**

[1] Tianjin Key Lab for Advanced Signal Processing, Civil Aviation University of China, Tianjin 300300, China

[2] Engineering Techniques Training Center, Civil Aviation University of China, Tianjin 300300, China

\* Correspondence: phan@cauc.edu.cn

**Abstract:** Due to the unique imaging mechanism of synthetic aperture radar (SAR), which leads to a discrete state of aircraft targets in images, its detection performance is vulnerable to the influence of complex ground objects. Although existing deep learning detection algorithms show good performance, they generally use a feature pyramid neck design and large backbone network, which reduces the detection efficiency to some extent. To address these problems, we propose a simple and efficient attention network (SEAN) in this paper, which takes YOLOv5s as the baseline. First, we shallow the depth of the backbone network and introduce a structural re-parameterization technique to increase the feature extraction capability of the backbone. Second, the neck architecture is designed by using a residual dilated module (RDM), a low-level semantic enhancement module (LSEM), and a localization attention module (LAM), substantially reducing the number of parameters and computation of the network. The results on the Gaofen-3 aircraft target dataset show that this method achieves 97.7% AP at a speed of 83.3 FPS on a Tesla M60, exceeding YOLOv5s by 1.3% AP and 8.7 FPS with 40.51% of the parameters and 86.25% of the FLOPs.

**Keywords:** synthetic aperture radar (SAR); aircraft detection; attention mechanism; residual dilated; structural re-parameterization

## 1. Introduction

Synthetic aperture radar (SAR) is an active microwave imaging sensor with all-day and all-weather capabilities; it is widely used in natural disaster monitoring, military reconnaissance, and urban planning [1,2]. As a typical target, aircraft have essential value in both military and civilian fields. Therefore, SAR aircraft interpretation has always been one of the research hotspots. With the continuous improvement of SAR imaging resolution, there are higher requirements for the accuracy and speed of aircraft target detection.

Traditional SAR target detection methods mainly include the constant false alarm rate (CFAR) algorithm, based on the statistical distribution of background clutter [3,4], and the algorithm of manually extracted image features [5,6]. The CFAR algorithm is primarily influenced by the statistical characteristics of background clutter, and it is difficult to accurately model the scene of complex objects, which generates more false alarms. The algorithm for manual extraction of image features divides detection into two separate processes: feature extraction and target classification. However, the process of manually extracting features is complex and highly dependent on parameter settings. Thus robustness and generalization are difficult to guarantee.

With the rapid development of deep learning methods and the increase of high-resolution SAR data, the emergence of convolutional neural network (CNNs) has brought many breakthroughs in tasks such as SAR image segmentation [7], land-cover classification [8], and target detection [9]. Data-driven models have good robustness and generalization. At the same time, this method can actively extract high-level features, avoiding the complicated work

of manual feature selection in traditional algorithms. For the SAR aircraft detection task, He et al. [10] explored the positional relationship between the two discrete components of the head and tail of the aircraft target, detected the two components respectively, and then used the KNN algorithm to match the discrete components. Guo et al. [11] first extracted airport areas and clustered these regions to find suspected aircraft targets, then enhanced the scattering information of these regions of interest and sent them to the network for training. Zhao et al. [12] introduced the attention mechanism and dilated convolution to improve the design of the neck of the feature pyramid network (FPN) [13] to improve the detection accuracy. Kang et al. [9] used an attention mechanism at the neck of the network based on the FCOS [14] network. They introduced a priori information of strong scattering points to the output of multiple-scale detection heads, which can effectively filter out false alarms.

Although the above deep learning-based SAR aircraft target detection algorithms have high detection accuracy, they usually use large backbone networks [9,11,12,15,16] such as ResNet 50/101 [17] or complex feature pyramid [9,11,12,14,18–22] neck design. These operations are complicated, and the expansibility is insufficient. An overly large network model not only has a high training time cost but also reduces the efficiency of aircraft target detection during inference due to excessive parameters and calculations. Compared with optical large-scale benchmark datasets such as Microsoft COCO [23] (328 K images with 80 categories), the SAR dataset is small, and the semantic information is relatively simple, such as SSDD [24] (1.16 K images with one category). Therefore, if a large network model is used in the SAR target detection task, it is necessary to load a pre-training model for good parameter initialization. Otherwise, there is severe over-fitting. In addition, for the problem of less SAR label data, there are currently studies such as semi-supervised learning [25] or generative adversarial [26] to expand the dataset to solve the problem effectively. Considering the unique imaging mechanism of SAR images, aircraft targets are in a discrete state in the image, and their detection performance is easily affected by multi-faceted structures such as covered bridges and airport buildings. To address the above problems, we propose a simple and efficient attention network (SEAN) for aircraft target detection in SAR images in this paper. First, a shallow and efficient backbone network is designed by introducing a structural re-parameterization technique [27]. Second, combining the unique scattering mechanism of SAR images with the a priori information of aircraft, we design a simple network neck with no complex lateral connections and multi-scale structure. This structure utilizes the dilated convolution of residual connections combined with the shallow semantic information enhanced by the attention mechanism and can perform aircraft target detection well with only one scale of feature resolution.

The main contributions of our work in this paper can be summarized as follows:

(1) The proposed SEAN is a SAR aircraft target detection network with a simple structure, high accuracy, and high speed. Compared with the typical target detection algorithms from recent years, SEAN has apparent advantages in detection accuracy and speed on the Gaofen-3 dataset.

(2) An appropriate network size is selected to balance the detection accuracy and speed. Then, the backbone network's depth is explored, proving that the C4 feature of the backbone network is more suitable for aircraft target detection. Furthermore, this paper uses a structural re-parameterization technique on the shallowed backbone to effectively enhance the feature extraction capability.

(3) A simple and efficient neck of the network is designed, discarding the complex feature pyramid network design. It mainly consists of three modules. One is a residual dilated module (RDM) that integrates the multi-scale receptive field. The second is a low-level semantic enhancement module (LSEM) that enhances the scattered information of SAR images. Furthermore, the third is a location attention module (LAM) that refines the multi-feature information after fusion.

## 2. Related Work

### 2.1. Deep Learning-Based Object Detection Algorithm

Object detection algorithms based on deep learning can be divided into two-stage [28,29] and one-stage [14,30,31] according to the number of detection stages. A two-stage detection algorithm has two detection heads. The first stage detects the candidate regions where there may be objects, and the second stage predicts the category and position of the object in the candidate regions. The two-stage detection algorithm represented by Faster-RCNN [29] usually has high detection accuracy. However, more complex structures have poorer detection speed than one-stage methods. The one-stage detection algorithm represented by YOLO [32] directly predicts the category and position of the object. This end-to-end method usually has faster detection speed and good detection accuracy. To obtain better feature extraction ability on large benchmark datasets such as ImageNet, a large backbone network such as ResNet101 is usually used. Recently, Vision Transformer (ViT) [33] as a backbone network has shown better feature extraction ability than CNN, which performs well on various visual downstream tasks [34]. However, this work shows that the performance of ViT exceeds that of CNN on the premise that there are hundreds of millions of data for pre-training. Otherwise, ViT's performance is not as good as CNN on datasets below a million. Therefore, in SAR aircraft detection with a small amount of data, CNN-based detection algorithms are mainly used for research [10–12]. Luo et al. [20] showed that the CSPDarknet backbone of the YOLO series is less affected by background interference, and it is more suitable for SAR aircraft target detection tasks than backbones such as ResNet.

For a long time, the YOLO series of algorithms [30–32,35–37] have always pursued the best balance between accuracy and speed, and the method in this paper is an improvement based on YOLOv5 6.0 [37] . YOLOv5 provides a total of five model sizes of n, s, m, l, and x according to the different widths and depths of the network. As shown in Figure 1, the network architecture of YOLOv5 6.0 can be divided into three parts: backbone, neck, and head. The backbone is an optimized CSPDarknet that has five stages for feature extraction and generates five scale features, and then sends the C3, C4, and C5 feature maps output by the last three stages to the neck. The neck is the optimized PANet [38], which has two branches for top-down and bottom-up multi-scale feature fusion. The head follows the coupled head of YOLOv3 [30] and detects objects of large, medium, and small scales based on nine preset anchor boxes. The loss function of the algorithm is composed of BCE as classification loss and objectness loss, and CIoU [39] as regression loss for bounding box prediction.

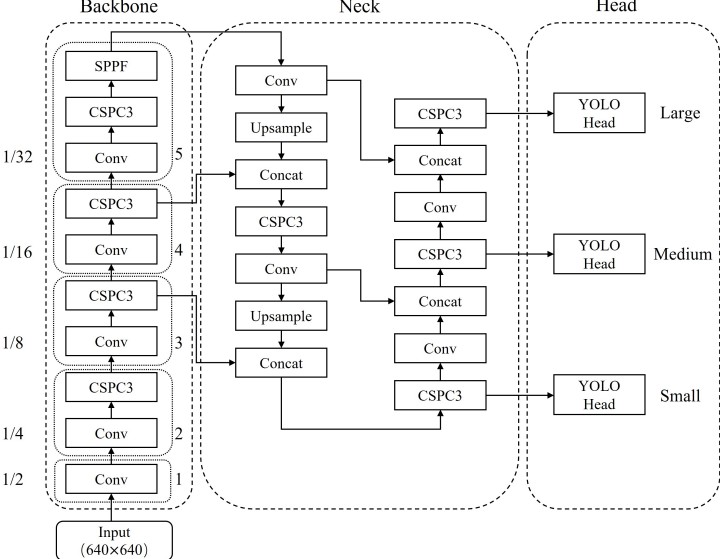

**Figure 1.** The architecture of YOLOv5 version 6.0.

### 2.2. Structure of the Detector Neck

In the typical object detection algorithms in recent years, it is indispensable to use the feature pyramid network (FPN) [13] as the neck of the network. Early CNNs [32] only used the C5 feature for detection, and its performance for small target detection was generally not high. Since then, the proposed method of top-down delivery of higher-order semantic information by FPN has effectively solved the low performance of small target detection. FPN usually brings two benefits: (1) performs multi-scale feature fusion, and (2) assigns objects of different scales to different receptive field features for detection. However, people often attribute the effectiveness of FPN to its ability to perform multi-scale feature fusion. With the success of FPN, some effective improvement works have appeared one after another. PANet [38] adds a bottom-up line based on FPN, which can pass up the low-level semantic information for spatial localization for fusion. Subsequently, there have been some works on complex laterally connected feature pyramids, such as BiFPN [40] and NAS-FPN [41]. These aim to obtain good detection accuracy through better multi-scale feature fusion. Recent SAR aircraft target detection algorithms [9,11,12,14,18–22] use an FPN as the neck. They usually introduce attention mechanisms [9,11,12,18] in the lateral connections to enhance the learning of channel and spatial information.

However, these FPN methods impose a large memory and computational burden, resulting in low efficiency for object detection. A recent work, YOLOF [42] showed that the significant effect brought by FPN mainly solves the optimization problem of multi-scale object detection in a divide-and-conquer way and does not rely on the part of multi-scale feature fusion. Moreover, it proposes a simple neck design with a label-matching mechanism that can have excellent detection accuracy even with only the top C5 feature, but its detection performance for small objects is poor compared to that of YOLOv4 [35]. Such a finding is not accidental, and some recent work on object detection in ViT has also shown the non-essentiality of complex lateral connections. ViTDet [34] showed that operation without lateral connections achieved good performance only using the topmost feature by simply constructing a pyramid structure by downsampling and up-sampling. In addition, AdaMixer [43] designed an adaptive sampling decoder to replace the FPN that can converge quickly and has good detection performance. It is worth mentioning that when AdaMixer did the ablation experiment of the backbone, it was found that the accuracy of ResNet only using the C4 feature for detection was better than that of the C3 and C5 features, which is similar to the conclusion of the work on the shallowed backbone in this paper.

### 2.3. Attention Mechanism

In recent years, attention mechanisms have emerged to allow computer vision systems to mimic the human visual system and find salient regions in complex scenes naturally and efficiently. The attention mechanism in computer vision can be regarded as the process of dynamically adjusting the weights based on the features of the input image [44]. This mechanism is used as a plug-and-play module, which can exchange high performance with a small amount of computational overhead and is now widely used in various computer vision tasks.

The detection network usually adds attention mechanisms such as SE [45] and CBAM [46] to enhance performance. However, SE only considers the connection between internal channel information and ignores the importance of spatial information. At the same time, CBAM introduces the local pooling operation of spatial information to the channel, but it suffers from the inability to obtain a large range of dependent information. The proposed coordinate attention mechanism (CoordAtt) [47] alleviates the above problems by embedding large-scale location information into channel attention. The structure of CoordAtt is shown in Figure 2 and consists of two main steps: coordinate information embedding and coordinate attention generation. In the first step, the input feature map X is first aggregated into two separate orientation-aware feature maps along the horizontal and vertical directions through two one-dimensional global pooling operations. In the second

step, the two feature maps are first spliced with embedded specific orientation information, and then, a 1 × 1 convolution and sigmoid function is used to output an intermediate feature map encoded with horizontal and vertical spatial information. The intermediate feature map is then split into two separate vectors along the spatial dimension, the channels are raised to the same dimension as the input feature map through two 1 × 1 convolutions, and the attention maps are output by the sigmoid function as $g^h$ and $g^w$. Each attention map captures long-range dependencies of input feature maps along one spatial direction, and location information can thus be preserved in the generated attention map. For example, Formula (1) calculates the weight of $y_c$ of the cth channel of the output:

$$y_c(i,j) = x_c(i,j) \times g_c^h(i) \times g_c^w(j) \tag{1}$$

The representation of attention regions can be emphasized by applying both attention maps to the input feature map by multiplication.

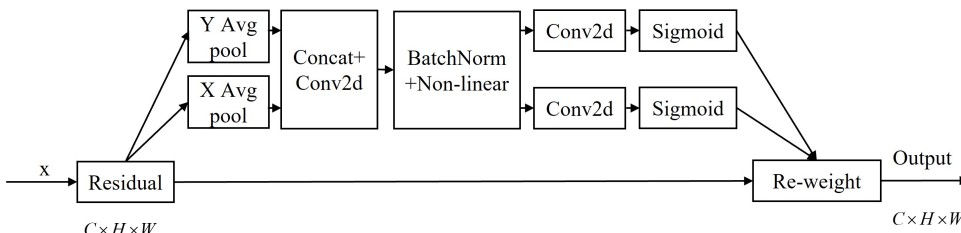

**Figure 2.** Coordinate Attention (CoordAtt).

## 3. Methodology

### 3.1. Overview of the Architecture of the Proposed SEAN

In order to detect aircraft accurately and quickly in SAR images with complex background interference, we propose a simple and efficient attention network (SEAN) architecture, as shown in Figure 3. Considering the trade-off between speed and accuracy, this paper selects YOLOv5s version 6.0 [37] as our baseline and makes a series of improvements for the SAR aircraft target detection task. The detailed network configuration of SEAN is shown in Table 1. In the table, n represents the stacked times of the module, and arguments represent the parameter information of the module, including input channel, output channel, kernel size, stride, and padding.

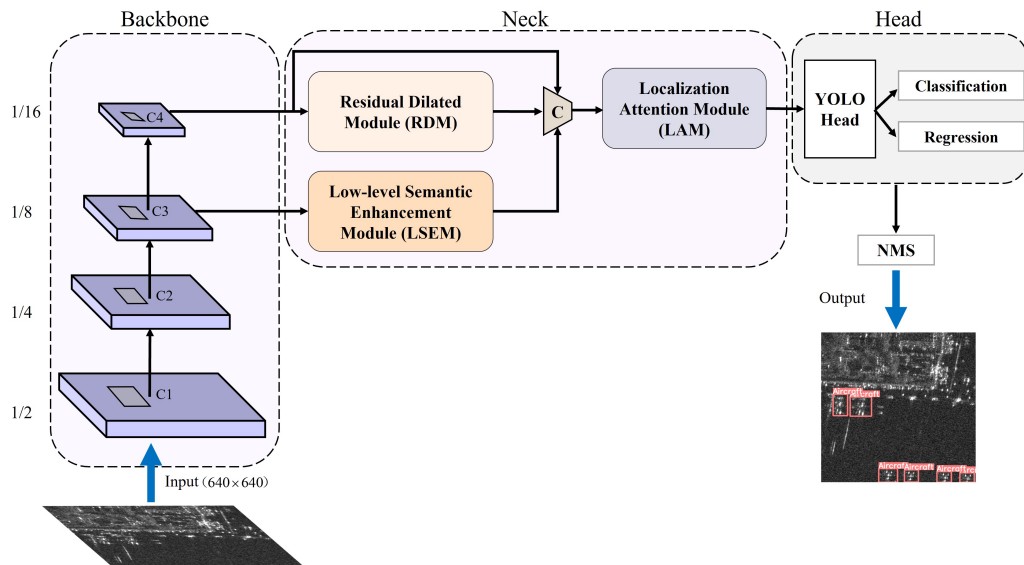

**Figure 3.** The overall architecture of the proposed SEAN. The network includes the backbone, the neck, and the head.

The backbone's primary function is to obtain good feature extraction ability. In this paper, the depth of the backbone network is preliminarily explored for the SAR aircraft target detection task, and structural re-parameterization technology [27,48] is introduced to enhance its feature extraction ability. For the neck, different from the usual complex lateral connection design to achieve multi-scale detection and multi-feature fusion, this paper designs a simple neck consisting of three parts: a residual dilated module (RDM), a low-level semantic enhancement Module (LSEM), and a localization attention module (LAM). The following is a detailed introduction to the method in this paper.

**Table 1.** Network configuration of the proposed SEAN.

|  | Sequence | From | n | Module | Arguments |
|---|---|---|---|---|---|
| Backbone | 1 | −1 | 1 | Conv | [3, 32, 6, 2, 2] |
|  | 2 | −1 | 1 | RepConv | [32, 64, 3, 2] |
|  | 3 | −1 | 1 | CSPC3 | [64, 64] |
|  | 4 | −1 | 1 | RepConv | [64, 128] |
|  | 5 | −1 | 2 | CSPC3 | [128, 128] |
|  | 6 | −1 | 1 | RepConv | [128, 256, 3, 2] |
|  | 7 | −1 | 3 | CSPC3 | [256, 256] |
| Neck | 8 | −1 | 1 | RDM | [256, 256] |
|  | 9 | 5 | 1 | LSEM | [128, 128] |
|  | 10 | −1 | 1 | RepConv | [128, 256, 3, 2] |
|  | 11 | [−1, 7, 8] | 1 | Concat | [1] |
|  | 12 | −1 | 1 | LAM | [756, 512] |
| Head | 13 | −1 | 1 | YOLOHead | [512, 1] |

### 3.2. Optimization of Backbone Network

SAR images are presented as single-channel grayscale images, which contain far less semantic information than the optical benchmark dataset. Moreover, aircraft are mainly displayed in the images as discrete strong scattering points, which are concentrated in the low-level semantic information. From the perspective of the receptive field, aircraft are usually a small target [49] in SAR images, and it is not necessary to use a deeper backbone to obtain a larger receptive field. In accordance with the above data characteristics, this section explores the network depth settings and introduces a structural re-parameterization technique to obtain better feature extraction capabilities.

### 3.2.1. Shallow Backbone

The backbone usually has five stages for hierarchical feature extraction. First, the resolution of the feature map of each higher stage is double downsampled, and the number of channels is doubled. Then, the last three stages of the backbone output C3, C4, and C5 features, which are used for multi-scale object detection. Since this paper does not use the complex FPN neck for target detection, it is necessary to explore at which stage the output feature map is more suitable for the aircraft detection task. Therefore, we make a preliminary exploration of the depth of the backbone. Note that the depth here refers to the number of stages selected. We use the C3, C4, and C5 output features with only one detection head for the aircraft detection task at a single resolution. We find that only the C4 feature map has good detection results, which may be attributed to the higher resolution of the C4 feature map being more suitable for medium and small aircraft detection tasks. Therefore, this paper uses the backbone without the fifth stage as the basis for subsequent work.

### 3.2.2. Backbone Re-Parameterization

Before each stage of the backbone performs feature extraction, the input feature map is downsampled to reduce the computational load of the subsequent network. However, downsampling reduces the resolution and inevitably loses some detailed information,

which is not conducive to positioning aircraft targets. As shown in Figure 4a, the basic Conv block is downsampled by convolution with kernel 3 and stride 2, followed by the BN layer and SiLU activation function to maintain better gradient transfer. To better retain feature information during downsampling, this paper introduces a structural re-parameterization technique of RepVGG [27] to optimize the Conv block. The core idea of the structural re-parameterization technique is that the network realizes the equivalent conversion of the structure through the equivalent conversion of parameters. Specifically, a set of branches with a convolution kernel size of 1 are added to the original Conv block during model training. The RepConv block is formed to increase the diversity of downsampling, and the parameters of the added branch are fused into the Conv block using the re-parameterization technique during inference so that the structure of the inference is the same as that of the Conv block, which enhances the capability of the model without affecting the detection speed. Convolution in deep learning is a cross-correlation operation. Since this operation is linear, the parameters of the convolution kernel increase with additivity. Figure 4b shows the simplified RepConv re-parameterization process. During inference, firstly, the BN layer is just a simple linear mapping, and its parameters can be directly added to the convolution kernel to improve the inference speed. It can be assumed that the number of channels in the input feature map X is 1, and the number of channels after downsampling by RepConv is 2. Then, use $A \in \mathbb{R}^{1 \times 2 \times 3 \times 3}$ and $B \in \mathbb{R}^{1 \times 2 \times 1 \times 1}$ to represent the weight parameters of the $3 \times 3$ and $1 \times 1$ convolutional layers, respectively. For a $1 \times 1$ convolution kernel, it can be equated to a special $3 \times 3$ convolution kernel with only non-zero values at the center. So that the weights of the convolution layers of the two branches can be summed up by channels and the summed weights can be denoted as $C \in \mathbb{R}^{1 \times 2 \times 3 \times 3}$, this paper uses "*" to represent the convolution operation; then, this process can be represented by Formula (2):

$$X * A + X * B = X * (A + B) \overset{A+B=C}{\Rightarrow} X * A + X * B = X * C \qquad (2)$$

**Figure 4.** Structural re-parameterization Conv block. K and S represent the size and stride of the convolution kernel. (**a**) Structural change process. (**b**) Re-parameterization process.

### 3.3. Neck Design of SEAN

FPN [13] is proposed to use a divide-and-conquer approach for multi-scale target detection, which alleviates the complex detection problem of small target objects. However, this complex lateral connection approach brings a significant memory and computation burden, which reduces detection efficiency. In this section, we wish to explore a simple and efficient network of necks to ensure accurate and fast SAR aircraft detection. We use the coordinate attention mechanism (CoordAtt) [47] as the base component of the neck to better capture the spatial localization information of the aircraft. In the following subsections, the three modules that make up the neck are described individually.

#### 3.3.1. Residual Dilated Module (RDM)

Considering that too much downsampling brings a larger receptive field, it also loses many details required for small target detection, and we hope to perform the detection task well on a feature map with only one resolution size; the dilated encoder approach [42] is used to alleviate these problems. We use dilated convolution after the optimized backbone

to increase the receptive field while maintaining the original feature map size. As shown in Figure 5, we use the Conv block to design a residual dilated module (RDM), which is formed by stacking two residual blocks with different expansion rates. The C4 feature map is the input to this module, which first enters the $1 \times 1$ convolution to reduce the channel dimension. Then, we use the dilated convolution to expand the receptive field, and we finally revert to the original channel number and fuse it with the input feature with the smaller receptive field. Figure 6 indicates the size of the aircraft targets covered by the receptive field in the feature map: (a) indicates that the receptive field of the C4 feature can cover most of the small- and medium-sized targets; (b) indicates the receptive field after dilated convolution can cover large- and medium-sized targets; and (c) indicates that the features of the two receptive fields can be fused to cover almost all target sizes through the RDM.

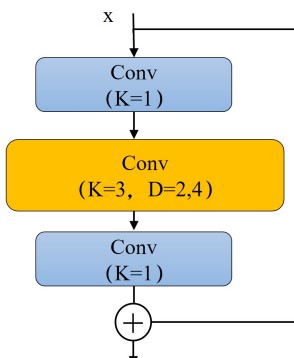

**Figure 5.** The structure of the RDM. K and D represent the size and dilation of the convolution kernel.

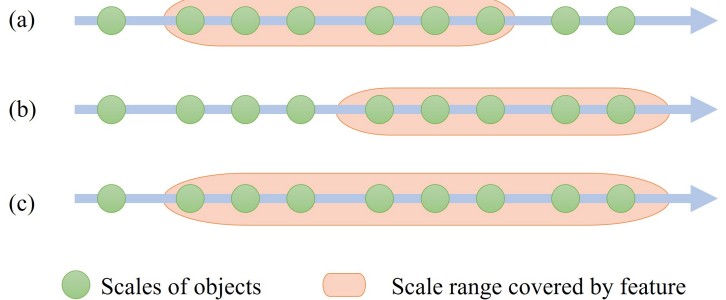

**Figure 6.** Size of aircraft covered by feature information. (**a**) The receptive field of the C4 feature. (**b**) The receptive field after dilated convolution. (**c**) The receptive field after the RDM.

### 3.3.2. Low-Level Semantic Enhancement Module (LSEM)

In SAR images, aircraft usually appear near complex objects with strong scattering points, such as covered bridges and buildings, which cause great interference to aircraft detection. We hope to better utilize low-level semantic information to suppress this interference. Unfortunately, the existing CNN algorithms mainly extract high-level semantic features of objects. By contrast, the high-level semantic features of SAR images are far less abundant than those of optical images. In order to better learn the spatial location information and scattering characteristics of SAR aircraft targets, this paper draws on the idea of inception [50] and introduces the coordinate attention mechanism (CoordAtt) [47] to design a low-level semantic enhancement module (LSEM). The specific structure of the LSEM is shown in Figure 7. The Conv block is used as the fundamental component of design. First, the C3 feature map is used as the input to reduce the dimension through the Conv block with the convolution kernel of 1 on the four branches, and then feature extraction is performed in different ways for each branch. Finally, the output features of the four branches are stitched to achieve the effect of multi-feature fusion of low-level semantic information.

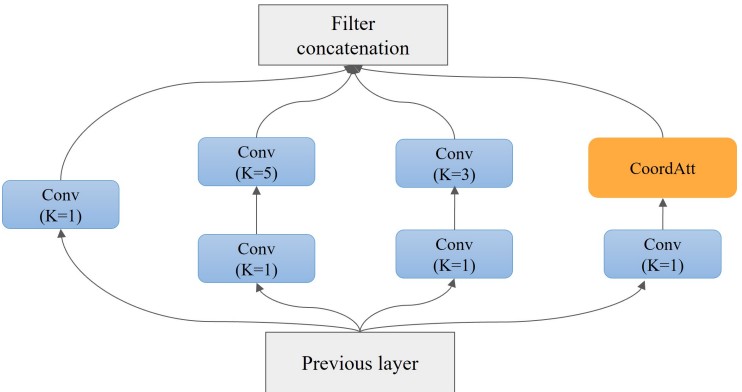

**Figure 7.** The structure of the LSEM. K is the size of the convolution kernel.

### 3.3.3. Localization Attention Module (LAM)

The fused feature map of the RDM and the LSEM already has enough information for aircraft target detection in complex environments. However, the feature information is redundant and easily confused due to surrounding objects. To this end, we design a localization attention module (LAM) to refine the fused feature map, hoping to achieve the effect that aircraft targets differ significantly from the surrounding features on different feature channels. As shown in Figure 8, the module first takes the fused output feature X as input through two branches. A branch first reduces the dimension of the input channel through a $1 \times 1$ Conv. It then refines the semantic features through a bottleneck block to perform a splicing operation with the other branch's $1 \times 1$ Conv channel dimension-reduction feature. The bottleneck block is a residual block formed by stacking two Conv blocks. Then, the output features are enhanced with spatial localization information by CoordAtt. Finally, the channel dimension and the nonlinearity of the network are maintained by a Conv block. In this way, LAM refines the features, strengthens the target's response on the feature map, and suppresses interference due to surrounding objects.

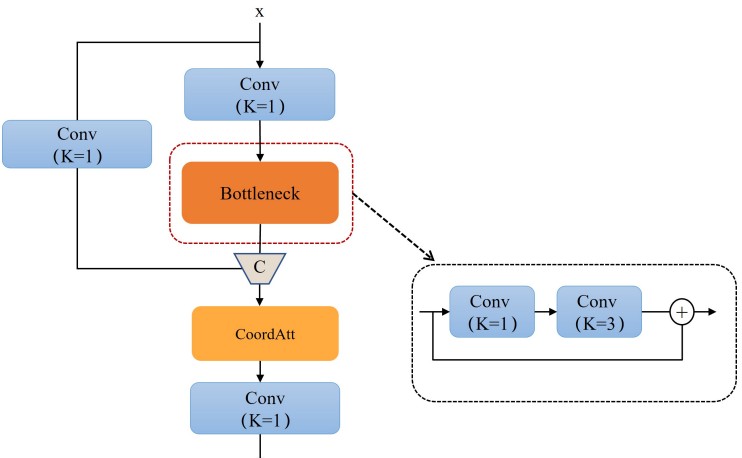

**Figure 8.** The structure of the LAM. K is the size of the convolution kernel.

## 4. Experiments and Analysis

### 4.1. Dataset Description

The Gaofen-3 aircraft target dataset [51,52] in this paper has been collected by the Gaofen-3 satellite and consists of single-polarization SAR image with a resolution of 1 m in the C-band and includes multi-temporal phase maps of multiple airports. The dataset has a total of 2000 image slices, with image sizes ranging from 600 to 2048 pixels. It mainly includes seven types of civil aircraft, such as the Boeing 737, with a total of 6556 aircraft samples. Figure 9 shows from the histogram of the bounding box distribution that there is only one aircraft in many images, but at most there are 35 aircraft in a picture. Figure 10 shows a partial

dataset slice, with the area marked by the green box the aircraft target. Furthermore, strong scattering points such as covered bridges and buildings are around the aircraft. By combining Figures 9–11, it can be seen that the aircraft targets in this dataset have uneven image quality, significant differences in the sizes of aircraft targets, dense target arrangement, and complex surrounding ground objects. This paper randomly divides the dataset into training, validation, and test sets at a ratio of 6:2:2 for experiments.

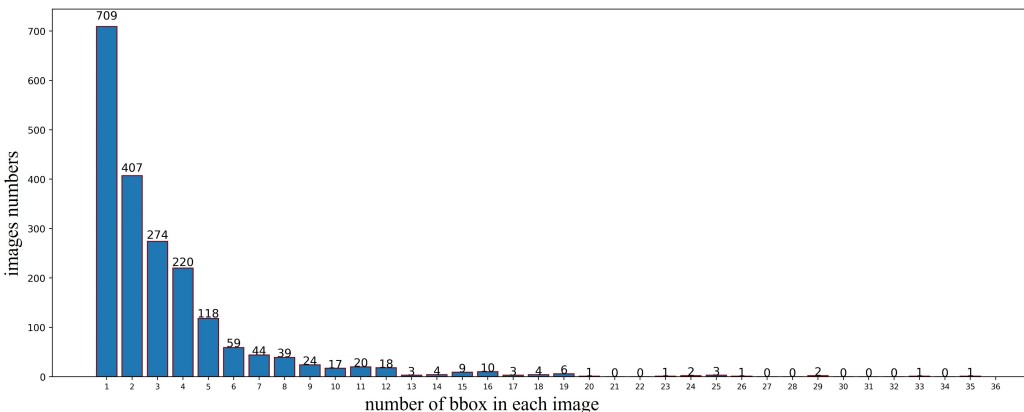

**Figure 9.** Distribution of the number of bounding boxes per image.

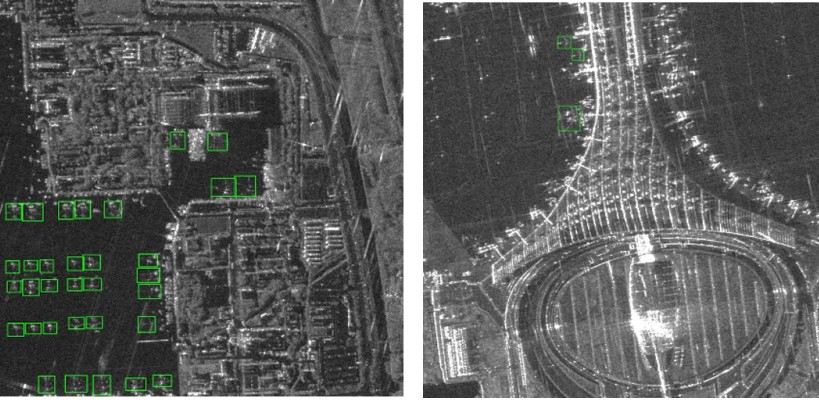

**Figure 10.** Part of the training samples. The green boxes represent the ground truth of the aircraft samples.

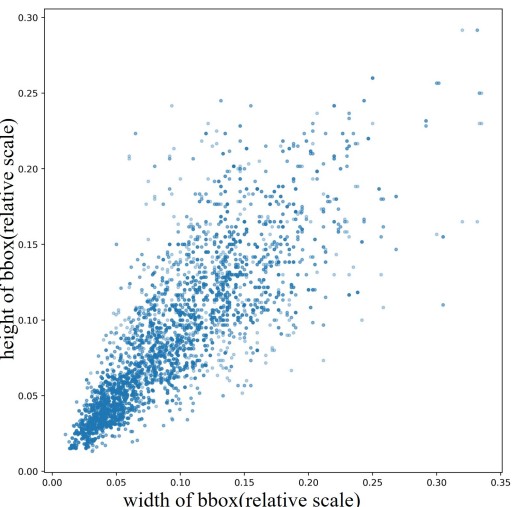

**Figure 11.** Relative size distribution of the bounding boxes. Dark blue dots indicate the presence of multiple aircraft targets of the same size.

### 4.2. Experimental Parameter Settings

The training image size is 640 × 640 pixels, and simple data augmentations such as panning, scaling, cropping, and flipping are done on the samples before training to enhance the model's generalization. Four groups of anchor boxes are preset as: (16, 14), (27, 25), (52, 50), (90, 83). Due to the corresponding changes to the backbone network in this paper, the network weights in all ablation experiments are initialized randomly. The optimizer for model training is SGD, with the momentum factor size of 0.937, the initial learning rate of 0.01, and the weight decay of 0.0005. We perform a learning rate warm-up in the first three epochs to maintain a better gradient. Each experiment is performed for 300 epochs, and the model with the best evaluation on the validation set is reserved for the final training result. All experiments are carried out under the Pytorch1.9 deep learning framework on an Ubuntu 16.04 system with two Tesla M60 (16 GB) GPUs.

### 4.3. Evaluation Metrics

In order to better evaluate the performance of each algorithm on aircraft target detection, we adopt some common evaluation metrics for object-detection tasks [9,53]: precision (P), recall (R), F1 score, and average precision (AP) to measure the detection performance of the algorithm; and model parameters (Params), floating-point operations per second (FLOPs), and frames per second (FPS) to measure model complexity and inference speed. In the comparison algorithm, we also draw the precision–recall (PR) curve to show the detection performance of each algorithm.

IoU = 0.5 is used as the threshold for dividing positive and negative samples in the experimental evaluation. Precision (P) and recall (R) are, respectively, defined in Formulas (3) and (4), where TP, FP, TN, and FN denote true case, false positive case, true negative case, and false negative case, respectively.

$$P = \frac{TP}{TP + FP} \tag{3}$$

$$R = \frac{TP}{TP + FN} \tag{4}$$

F1 score is based on the harmonic mean of P and R. The larger the value of the score, the better the model's performance. It is defined in Formula (5):

$$F1 = \frac{2 \times P \times R}{P + R} \tag{5}$$

Average precision (AP) is the area enclosed by the PR curve and the coordinate axis. It comprehensively considers the effects of precision (P) and recall (R) to reflect the quality of the model. The larger the AP value, the better the model performance. Its definition is Formula (6):

$$AP = \int_0^1 p(r)dr \tag{6}$$

Model parameters (Params) measure the space complexity of the model and its corresponding memory resource occupation. For example, assuming that the current convolution layer uses K, M, and N to represent the size of the convolution kernel, the number of input channels, and the number of output channels, respectively, then this convolutional layer parameter quantity is defined in Formula (7):

$$Params = K^2 \times M \times N \tag{7}$$

Floating-point operations per second (FLOPs) measures the time complexity of the model, which is a reference indicator of the calculation time. Assuming that the current convolutional layer is represented by K, M, N, H, and W for the convolutional kernel size,

input channels, output channels, and the height and width of the output feature map, respectively, the FLOPs of this convolutional layer is defined in Formula (8):

$$FLOP s = 2K^2 \times M \times N \times H \times W \tag{8}$$

Frames per second (FPS) measures the overall detection speed of the algorithm and is defined in Formula (9), which gives the average detection time of an image. A larger value represents faster detection.

$$FPS = \frac{1}{t} \tag{9}$$

*4.4. Experimental Results and Analysis*

4.4.1. Selection of Preset Anchor Boxes

Since the method in this paper does not use FPN as the neck and only detects at one resolution of 16-times downsampling, if the original scale of the anchor box preset is used, anchor box recall on the dataset is low, and there are many missed detections. Therefore, it is necessary to redesign a set of anchor boxes based on the dataset's labeling information to optimize the model's performance. Figure 11 shows the aspect ratio of the aircraft bounding boxes relative to the images; it can be seen that the aspect ratio of the aircraft target is close to 1:1, and most of the aircraft are small relative to the image. From a relative scale perspective, a target is usually defined as small when the ratio of the target's bounding box area to the image area is less than 0.58% [49]. Data analysis finds that small targets account for 48.8% of the SAR aircraft dataset. For this reason, we determine the bounding box size of the training samples by the K-means clustering method and finally find four groups of anchor-box presets suitable for aircraft samples: (16, 14), (27, 25), (52, 50), (90, 83).

4.4.2. Ablation Experiments and Analysis

This section is a series of ablation experiments on the SEAN network architecture proposed in this paper to evaluate the effectiveness of each module based on evaluation metrics. The experiments are divided into three parts: choice of model size and depth, module effectiveness, and comparison between different backbone networks and necks on YOLOv5s.

(1) Selection of Model Size and Depth: This paper conducts experiments using each of the five model sizes provided by YOLOv5 version 6.0. As shown in Table 2, YOLOv5s has the best trade-off between accuracy and speed, and a short training time (T-time). Secondly, this paper explores the depth of the model, that is, which stage output features are more suitable for SAR aircraft target detection. As shown in Table 3, detection using only the C4 feature has a sufficiently high AP of 95.4%, which is 14.3% and 1.1% higher than that of the C3 and C5 features, indicating that the C4 feature has a good trade-off between resolution and high-level semantic information. On the other hand, the FPS is low when only the C3 feature is used for detection, although the FLOPs are minimal. This is due to the lack of high-level semantic information in the C3 feature map for the binary classification of ground objects and aircraft targets, which results in many false alarms in the prediction. This leads to short network inference time when using only the C3 features for detection, but too many false alarms are generated and too much time is spent on NMS post-processing of the anchor boxes. Based on the above experimental analysis, we select the C4 feature map of YOLOv5s plus the head as the basis for subsequent experiments.

(2) Effectiveness of Each Module: In order to clearly express the contribution of each module of SEAN, we conduct experiments based on the C4 plus the head. The results are shown in Table 4. Firstly, the structural re-parameterization technique is introduced into the backbone, i.e., RepConv is used instead of simple convolutional downsampling. It can be seen that the model does not affect the detection FPS, and the AP is improved by 0.6%. Then, for the neck design, we use RDM to fuse different receptive-field scale features to achieve 0.3% AP improvement, and LSEM to enhance the learning of low-level semantic information to achieve 0.4% AP improvement. Since the semantic information of directly

splicing the output features of both RDM and LSEM is more confusing and redundant, LAM is added in this paper to refine the semantic information, which improves the AP by 1% by enhancing aircraft positioning information. Moreover, the AP of the loss function of SIoU [54] using the frontier later decreases by 0.4%. Figure 12 shows (a) the actual detection map of the method in this paper; (b) in the absence of LAM, the aircraft target is easily confused with the surrounding features on these grayscale channel maps; and (c) with LAM, the corresponding performance of the aircraft target on the grayscale channel maps is significantly different from that of the surrounding features. It can be seen that LAM can refine the semantic features and effectively suppress the interference of complex features on aircraft target detection. The attention mechanism can compensate for the problem of strong locality and insufficient globality of CNN by obtaining global context information. Table 5 shows the effects of different attention mechanisms on the performance of LAM. We added different attention modules in the same position based on the LAM without attention mechanism to verify the effect. The results show that almost all of these attention mechanisms improve the performance of the network; specifically, the overall performance of our adopted CoordAtt is the best. In general, the algorithm in this paper significantly reduces the parameters and FLOPs compared to YOLOv5s and improves the AP by 1.3% and 8.7 FPS on the test set. Figure 13 shows the change to AP on the validation set with increasing epochs during model training. Since SEAN only uses one scale of detection head, the model fitting speed is slower than that of YOLOv5s, but the final AP value is better.

**Table 2.** Selection of model size.

| Method | P (%) | R (%) | F1 (%) | AP (%) | Params (M) | FLOPs (G) | T-Time (h) | FPS |
|---|---|---|---|---|---|---|---|---|
| YOLOv5n | 93.8 | 91.6 | 92.7 | 95.6 | **1.77** | **4.2** | **0.84** | **123.3** |
| YOLOv5s | 93.7 | 92.4 | 93.0 | 96.4 | 7.01 | 16.0 | 1.15 | 74.6 |
| YOLOv5m | 93.5 | 94.2 | 93.8 | 96.5 | 20.87 | 48.0 | 2.60 | 36.1 |
| YOLOv5l | **94.1** | 93.8 | 94.0 | 96.4 | 46.11 | 107.8 | 4.67 | 22.7 |
| YOLOv5x | 93.4 | 95.3 | **94.4** | **96.6** | 86.12 | 204.2 | 9.19 | 13.1 |

**Table 3.** Selection of backbone depth.

| Method | P (%) | R (%) | F1 (%) | AP (%) | Params (M) | FLOPs (G) | FPS |
|---|---|---|---|---|---|---|---|
| C3+Head | 81.8 | 72.4 | 76.8 | 81.1 | **0.23** | **5.1** | 58.8 |
| C4+Head | 93.3 | **92.1** | **92.7** | **95.4** | 1.15 | 8.0 | **107.5** |
| C5+Head | **94.4** | 87.3 | 90.7 | 94.3 | 3.52 | 9.9 | 104.2 |

**Table 4.** Ablation experiments of each SEAN module.

| Method | P (%) | R (%) | F1 (%) | AP (%) | Params (M) | FLOPs (G) | FPS |
|---|---|---|---|---|---|---|---|
| YOLOv5s | 93.7 | 92.4 | 93.0 | 96.4 | 7.01 | 16.0 | 74.6 |
| C4+Head | 93.3 | 92.1 | 92.7 | 95.4 | **1.15** | **8.0** | **107.5** |
| +RepConv | 91.6 | 94.1 | 92.8 | 96.0 (+0.6) | **1.15** | **8.0** | **107.5** |
| +RDM | 94.7 | 92.7 | 93.7 | 96.3 (+0.3) | 1.16 | 8.1 | 103.2 |
| +LSEM | 93.7 | 93.5 | 93.6 | 96.7 (+0.4) | 1.51 | 9.6 | 92.6 |
| +LAM (ours) | **95.6** | 94.2 | **94.9** | **97.7** (+1.0) | 2.84 | 13.8 | 83.3 |
| +SIoU [54] | 94.2 | **95.5** | **94.9** | 97.3 (−0.4) | 2.84 | 13.8 | 82.6 |

**Table 5.** The impact of different attention mechanisms on LAM.

| Method | P (%) | R (%) | F1 (%) | AP (%) |
|---|---|---|---|---|
| LAM (Basic) | 94.6 | 94.5 | 94.5 | 97.2 |
| LAM (SE [51]) | 94.9 | 94.4 | 94.6 | 97.3 |
| LAM (CBAM [54]) | 93.7 | 94.6 | 94.1 | 97.5 |
| LAM (TripletAtt [55]) | 93.9 | **95.1** | 94.5 | 97.2 |
| LAM (CoordAtt [47]) | **95.6** | 94.2 | **94.9** | **97.7** |

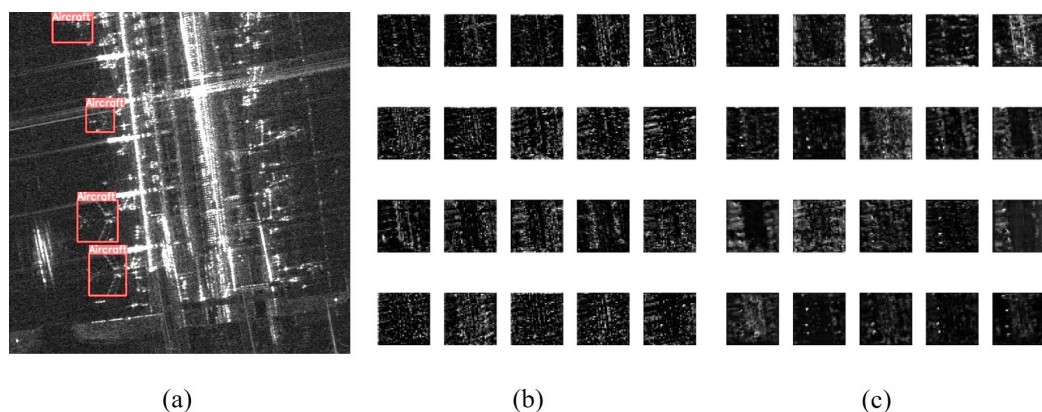

|     (a)     |     (b)     |     (c)     |

**Figure 12.** Channel feature visualization. (**a**) The actual detection map of the method in this paper. (**b**) Feature channel map without LAM. (**c**) Feature channel map with LAM.

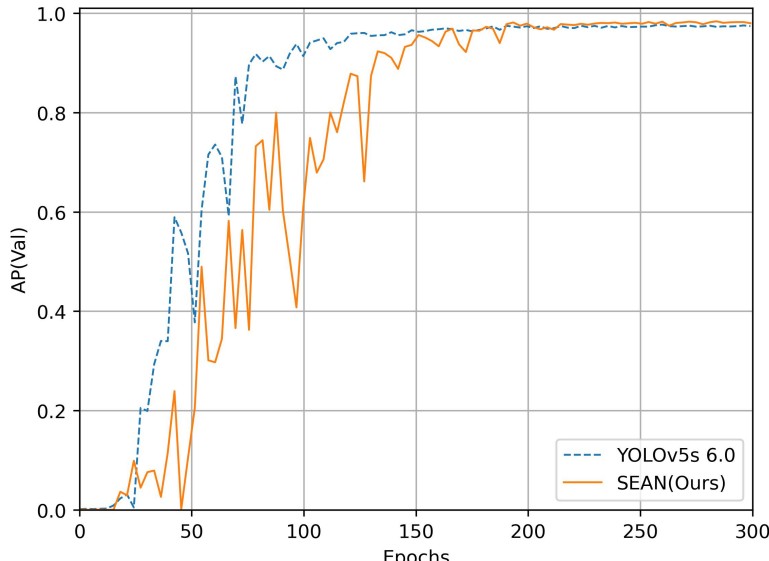

**Figure 13.** Training curves on the validation set. We evaluate the AP of each epoch for model training, and the results show that SEAN is better than YOLOv5s.

To better understand the points of interest of the SEAN model under complex ground conditions, this paper uses the Grad-CAM [56] technique for visualization. This is a back-propagation gradient of the information from the predicted anchor boxes, and the learning of the model is represented by the response of the parameters of the last convolution layer using a heat map. In Figure 14a–c represent three cases of difficulty detecting due to complex ground objects, dense aircraft arrangement, and small aircraft targets. The actual detection effect of the model is in the upper half, and the corresponding Grad-CAM feature visualization is in the lower half. The figure shows that the model can accurately detect the aircraft targets in all these scenarios. Furthermore, the heat map shows that the areas with aircraft targets are red. In contrast, other objects such as corridors, buildings, and runways are presented in cool colors, which proves that our algorithm has strong anti-interference ability and high detection accuracy in complex scenes.

(3) Different Neck and Backbone: To further demonstrate the performance advantage of the proposed method, we substitute the backbone and neck of YOLOv5s for comparison with SEAN. For the backbone, we use ConvNeXt [57], and we train its tiny version with the same number of channels per module control as with YOLOv5s. Furthermore, the neck part is changed to FPN [13] and BiFPN [40]. Table 6 shows that our proposed algorithm has better accuracy and speed than YOLOv5s using different backbones and necks.

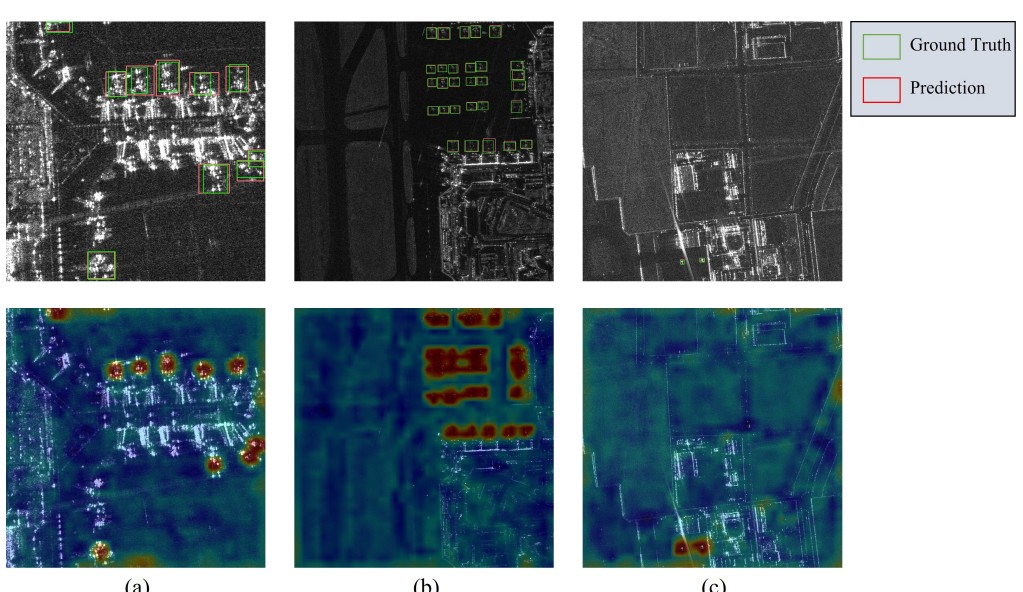

(a)          (b)          (c)

**Figure 14.** Grad-CAM visualization of three scenes (**a–c**); the upper part is the prediction result of the SEAN model, and the bottom part is the visualized heat map.

**Table 6.** Comparison to YOLOv5 with different backbones and necks.

| Method | Backbone | Neck | F1 (%) | AP (%) | Params (M) | FLOPs (G) | FPS |
|---|---|---|---|---|---|---|---|
| SEAN (ours) | Modified CSPDarknet | Ours | **94.9** | **97.7** | **2.84** | **13.8** | **83.3** |
| YOLOv5s | CSPDarknet | PANet | 93.0 | 96.4 | 7.01 | 16.0 | 74.6 |
|  | CSPDarknet | FPN | 92.7 | 96.0 | 6.85 | 15.8 | 78.2 |
|  | CSPDarknet | BiFPN | 93.6 | 96.5 | 7.07 | 16.1 | 74.2 |
|  | ConvNeXt [57] | PANet | 93.8 | 97.0 | 25.3 | 52.7 | 16.9 |

### 4.4.3. Comparative Experiments and Analysis

To further demonstrate the effectiveness of SEAN, we compare it to seven typical detection algorithms based on MMDetection [58]. As shown in Table 7, the long training time of relatively large backbone networks such as ResNet, especially two-stage algorithms, makes them prone to over-fitting on small-scale SAR data. Therefore, we use a pre-trained model for initialization in the backbone of these networks. In terms of detection speed, the table shows that models using ResNet50 as the backbone and FPN as the neck, such as Cascade R-CNN, have massive computational effort, resulting in significantly lower detection speeds. Moreover, detection speed is improved when YOLOF adopts the dilated encoder (D-en) as the neck. While YOLOv3, YOLOX-s, and SEAN (our method) all use a lightweight backbone, SEAN uses a simple neck to significantly improve detection speed. Regarding detection accuracy, YOLOv3, YOLOX-s, and SEAN (our method), which do not use ResNet50 as the backbone, are generally better than the other methods. As shown in Figure 15, the area of the PR curve of the SEAN algorithm is significantly larger than that of the other comparison algorithms, which means that the method in this paper has a significant detection accuracy advantage in actual detection. According to the above experiments, the proposed SEAN method has better average accuracy of 97.7%, a better F1 score of 94.9%, and faster detection speed of 83.3 FPS with fewer parameters and FLOPs than the typical algorithms on the SAR aircraft dataset.

In order to visualize the effectiveness of SEAN, we select some challenging detection scenarios from the test set to compare the algorithms. Figure 16 shows a small target scene, and SEAN accurately detects the small aircraft target. Figures 17 and 18 are two scenes with complex environments; SEAN has significantly fewer missed targets and false alarms compared to other models and can effectively avoid the influence of strong scattering points due to complex objects. Moreover, among the aircraft targets detected by SEAN, the

bounding boxes completely wrap whole aircraft, and there are no situations in which only local components are detected.

**Table 7.** Comparison to different algorithms.

| Method | Backbone | Neck | F1 (%) | AP (%) | Params (M) | FLOPs (G) | FPS |
|--------|----------|------|--------|--------|------------|-----------|-----|
| Faster R-CNN | ResNet50 | FPN | 88.9 | 86.1 | 41.12 | 182.3 | 11.2 |
| Cascade R-CNN | ResNet50 | FPN | 87.3 | 84.5 | 68.93 | 237.6 | 7.3 |
| FCOS | ResNet50 | FPN | 89.1 | 83.5 | 31.84 | 153.7 | 15.3 |
| YOLOF | ResNet50 | D-en | 83.8 | 79.4 | 42.06 | 78.5 | 22.3 |
| TOOD [59] | ResNet50 | FPN | 91.1 | 88.8 | 31.79 | 144.3 | 12.9 |
| YOLOv3 | MobileNetv2 [60] | FPN | 94.8 | 92.7 | 3.74 | **13.5** | 44.2 |
| YOLOX-s | CSPDarknet | PANet | 94.0 | 93.0 | 8.94 | 26.3 | 41.2 |
| SEAN (ours) | Modified CSPDarknet | Ours | **94.9** | **97.7** | **2.84** | 13.8 | **83.3** |

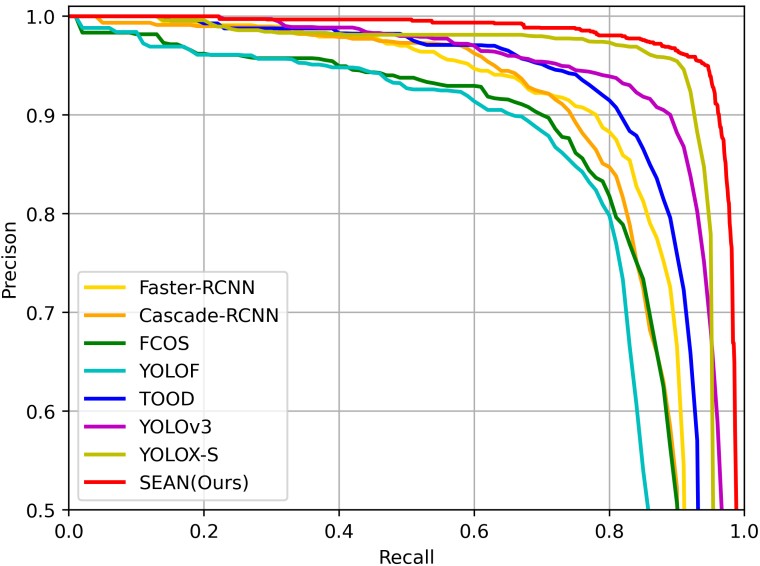

**Figure 15.** The PR curves of various methods.

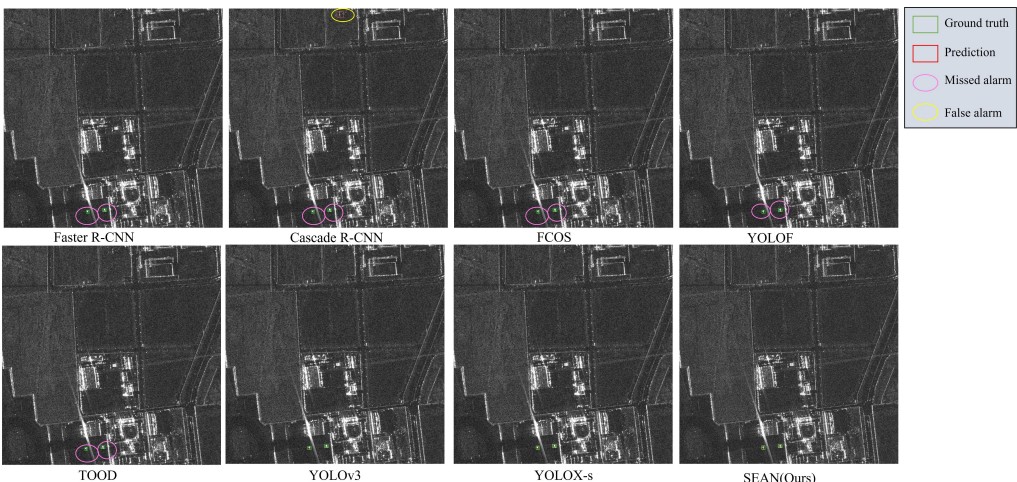

**Figure 16.** Scene 1. Small aircraft target detection.

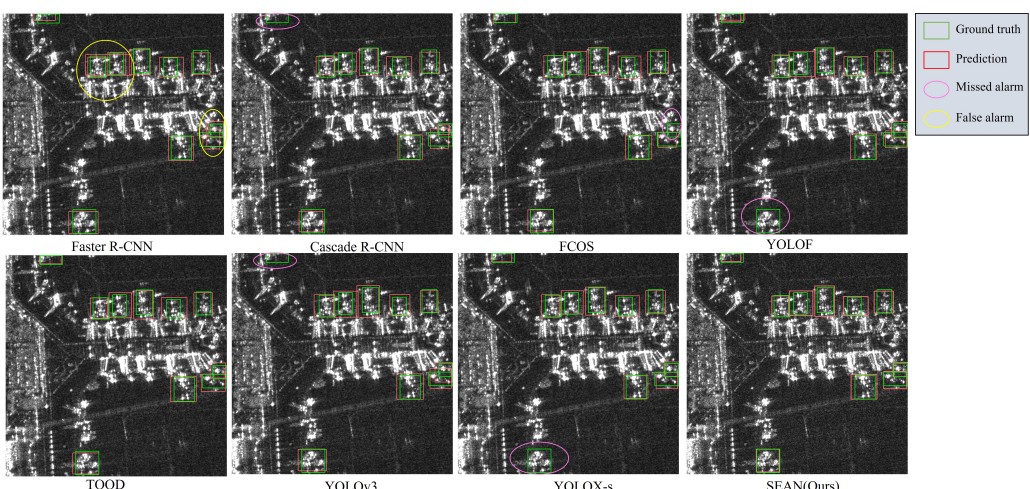

**Figure 17.** Scene 2. Aircraft target detection in complex environment.

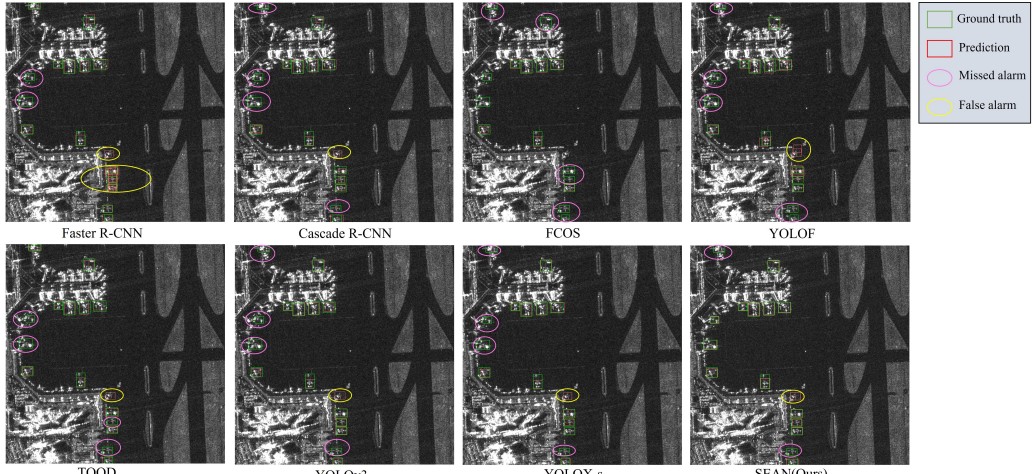

**Figure 18.** Scene 3. Aircraft target detection in complex environment.

## 5. Conclusions

In this paper, we propose a simple and efficient attention network (SEAN) for aircraft detection in SAR images, which avoids the previous deep backbone and complex, laterally connected FPN neck and improves the detection accuracy and speed while significantly reducing the parameters and the FLOPs in the network. Through experiments, SEAN achieves 97.7% AP and 83.3 FPS speed on the Gaofen-3 aircraft target dataset. The results show that this algorithm has apparent advantages in detection accuracy and speed for SAR aircraft targets in complex backgrounds compared to other typical target algorithms. It shows that the trained model with a large amount of SAR aircraft labeled data has a high enough detection accuracy. However, due to the difficulty of manual labeling of SAR aircraft targets and the small amount of SAR data, the detection of SAR aircraft with small samples and the use of self-supervised learning methods in the field of SAR is very promising. Furthermore, we also find that the C4 feature of the backbone is more suitable for aircraft detection, and it can also be used to conduct lightweight network research on SAR in the future.

**Author Contributions:** Methodology, P.H. and D.L.; Project administration, P.H. and Z.C.; Software, P.H. and D.L.; Validation, P.H., D.L., B.H. and Z.C.; Writing, P.H., D.L. and B.H. All authors have read and agreed to the published version of the manuscript.

**Funding:** This work was supported by the Central University Basic Scientific Research Project of China (no. 3122020043).

**Data Availability Statement:** Not applicable.

**Conflicts of Interest:** The authors declare no conflict of interest.

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
