# Peer review of "SEAN: A Simple and Efficient Attention Network for Aircraft Detection in SAR Images"

_remotesensing, doi:10.3390/rs14184669_

Round 1
Reviewer 1 Report
This paper proposes a new method to detect targets efficiently in SAR images. I have a few comments.
1. page 1, line 23, CFAR (Constant False Alarm Rate)
2. page 3, line 53, it is said that 'compared with the optical large-scale benchmark dataset, SAR dataset is small'. Please add some example dataset and its size to make it clear
3. page 2, line74, .. that the C4 features of the backbone network... I think it needs some additional explanation or reference since it's the first time that 'C4 features' is mentioned.
4. page 4, line 150, 'only using the C4 feature for detection .... is similar to the conclusion of the work in the shallowed backbone in this paper.' But, it seems that the backbone of this paper uses C3 as well as C4. So, it is a little bit confusing. Please make this clear.
5. page 4, line 161, it talks about SE[39] and CBAM[40]. It would be better if you add more recent previous attention network, if any, since both networks were published in 2018.
6. page 4, the end of line 167, 'The structure of CoordAtt is.
7. page 5, I think some explanation on the Head structure is necessary. For example, you can mention the difference between your Head and that of YOLOv5s.
8. page 6, session 3.2.2 Backbone re-parameterization. I think it would be better if you add an explanation of the original re-parameterization from Rep VGG. It is not easy to follow if the readers do not know re-parameterization.
9. page 9, line 296. 'The fused feature map of the RDM and the LSEM....' It is not clear how to fuse those two feature maps. Can you explain?
10. page 9, line 312. 'The experimental dataset[45,46] in this paper is...' Can you tell the exacnt name of the dataset?
11. page 13, line 405, '2) Effectiveness Analysis of Each module...' In order to find the effect of each module, it would be better to add each module one by one to the original YOLOv5s., not serially as in the paper. Is it possible?
12. page 14, line 428. 'To better understand the parts of ...' It is not clear why the authors added this paragraph. The advantage of the proposed method is summarized in the tables. There is no new information in this paragraph. Or you may compare the Gard-CAM feature of the original YOLOv5s here with yours.
Author Response
Dear Reviewer,
We want to thank you for the careful reviews, thoughtful and constructive comments, and valuable suggestions. We have made all the necessary amendments in our revised manuscript according to the comments and suggestions. All of the modified parts are highlighted in red in the revised manuscript. Please see the attachment for our reply to your concerns and comments.
Thank you for your time and consideration.
Best wishes,
Ping Han, Dayu Liao, Binbin Han, Zheng Cheng

Reviewer 2 Report
The authors present an exciting paper where they propose a simple and efficient attention network (SEAN), which takes YOLOv5s as the baseline. First, they shallow the depth of the backbone network and introduce a structural re-parameterization technique to increase the feature extraction capability of the backbone. Second, the neck architecture designed by using a residual dilated module (RDM), a low-level semantic enhancement module (LSEM) and a localization attention module (LAM) substantially reduces the number of parameters and computation of the network. The manuscript’s results are reproducible based on the details given in the methods section. The manuscript is well written and should greatly interest the readers. However, some figures could be more significant. Also, specify more about the pros and cons of YOLO5 instead of other YOLO versions. Finally, the conclusion should mention more about their future work.
Author Response

(The authors gave the same response as above.)

Reviewer 3 Report
This is a well-written paper, which study the issue of aircraft detection from SAR images using deep learning model with high detection accuracy and efficiency. My detailed comments are as follows:
1) Line 50, "These operations...the expansibility is insufficient". How about the expansibility of the proposed model? The authors can introduce more in Section 3.
2) Line 53,"SAR dataset is smal..., it will cause a severe over-fitting phenomenon." In fact, some studies have been presented to tackle the issue of limited SAR training dataset, such as the semi-supervised deep learning methods. The authors can introduced or cite related papers in this part.
3) The definitions of the symbols K, S, D in Fig. 4 and Fig. 5 should be given.
4) Line 319, "By Figure 11, it can be seen that the aircraft targets in this dataset have ... complex surrounding ground objects." Figure 11 only shows the different sizes of the aircraft, it is not enough to indicate the complex backgrounds of the images, so, consider to provide more information to indicate the above issue.
5) Has the speckle reduction approach been applied before training the model and detecting the targets? Does the speckle reduction approach have significant influence on the performance of the proposed method?
Author Response

(The authors gave the same response as above.)

Reviewer 4 Report
This paper introduces an aircraft detection method based on SEAN for SAR images. Taking YOLOv5 as the baseline, the backbone and neck design problems suitable for aircraft detection in SAR images are discussed in this paper. For the backbone part of detection framework, the C4 is selected as the main input feature layer, and structural re-parameterization technique is used to improve the feature extraction ability. Instead of feature pyramid network in the neck design part,this paper design a lightweight fusion network by residual dilated module, low-level semantic enhancement module and localization attention module. Through the above work, this paper achieves better detection results and faster detection speed with smaller model parameters and training time. However, there are still some problems in the paper to be improved as follows:
1: Keep Fig.1 and Fig.3 in the same pattern in order to show the difference between two algorithms .
2: There are some unnecessary grey lines in Figure 5, Figure 9, Figure 11 and Figure 12, please delete them.
3: The attention mechanism is common in target detection algorithms in recent years, and the original YOLOV5s does not use the attention mechanism. The ablation experiment in this paper only compares the Neck design of YOLOv5s, but does not compare it with other attention mechanisms. It would be better if comparison experiments are considered.
4: The English expression in this paper needs to be improved.
In general, this paper has done much innovative work, and it is suggested to be accepted after finishing revision the above problems.
Author Response

(The authors gave the same response as above.)

Reviewer 5 Report
Paper ID: remotesensing-1835343-peer-review-v1
SEAN: A Simple and Efficient Attention Network for Aircraft Detection in SAR Images
In this paper, the authors propose a simple and efficient attention network. The innovation is that the authors shallow the depth of the network and introduce a structural re-parameterization technique to increase the feature extraction capability. As a result, the proposed SEAN shows competitive detection accuracy and low model complexity. However, the novelty of the presentation should be improved, and there are some problems that need to be solved.
1. In the abstract, the authors mentioned “the feature pyramid neck design and large backbone network can reduce the detection efficiency to some extent”, and the SEAN was proposed to address that problem. Therefore, more detailed discussion of the superiority of the detection efficiency of the SEAN is needed.
2. As I understand it, Figure 13 represents that the detection accuracy of the SEAN is better than YOLOv5s. However, the convergence of the SEAN is worse than YOLOv5s. Can the authors explain the reasons for the decrease in convergence? Does the change to the network structure result in a decrease in the convergence?
3. In Table 4, it is confusing that only the precision and average precision of the proposed method are better than other methods.
4. In Table 6, the FLOPs representing the time complexity of the SEAN is higher than YOLOv3, which is related to the detection efficiency of the proposed method. Can the authors explain the reasons for the decrease in detection efficiency, since the FLOPs is one of the important metrics of the proposed method.
5. More explanation about Figure 11 is needed. For example, what is the meaning of the circles in dark blue and light blue, respectively?
6. It is recommended to use different line types to distinguish different methods in Figure 13 and 15.
7. Formulas in Section 4.3. are the common evaluation metrics. The citation should be added.
8. Could you give more explanations about the intention that you are to design a simple and efficient network? SAR image in amplitude is just a black and white image with 8 digits for quantification, which only includes limited information in terms of amplitude. Therefore, it is definite to design a simple network to cope with the SAR image data. I think there would be only a little improvement when you increase the complexity of the network.
9. More attention should be paid on the format and the written English of the manuscript.
Author Response

(The authors gave the same response as above.)

Round 2
Reviewer 1 Report
The authors addressed my previous comments well enough to publish.
Thanks.
Reviewer 3 Report
Conguratulations! The authors have done good work to improve the quality of this paper. All my questions are addressed.